

# Deep-water red shrimp fishery in the Eastern-Central Mediterranean Sea: AIS-observed monthly fishing effort and frequency over four years

Jacopo Pulcinella[1,2], Enrico Nicola Armelloni[1,3], Carmen Ferrà[1,3], Giuseppe Scarcella[1], Anna Nora
Tassetti[1]

[1]National Research Council, Institute of Marine Biological Resources and Biotechnologies (CNR-IRBIM), Ancona, Italy
[2]Almawave SpA, Rome, Italy
[3]Department of Biological, Geological and Environmental Sciences (BiGeA), University of Bologna, Bologna, Italy

*Correspondence to*: Enrico N. Armelloni (enrico.armelloni@irbim.cnr.it)

**Abstract.** Deep-sea fishery in the Mediterranean Sea was historically driven by the commercial profitability of deepwater red shrimps and understanding spatio-temporal dynamics of fishing is key to comprehensively evaluate the status of these profitable resources and prevent stock collapse. A four-year time series of AIS-based observed monthly patterns and related frequency of trawling disturbance are provided with a resolution of 0.01°*0.01°, accounting for the spatial extent and temporal variability in deep water bottom contact fisheries during the period 2015-2018. The dataset was estimated from 370 fishing vessels that were found to perform trawling in deep water (400 m - 800 m) during the study period, and they represent a significant part of the real fleet exploiting this fishing grounds in the study area. The reconstructed deep-water trawling effort dataset is available at: https://doi.org/10.17882/89150 (Pulcinella et al., 2022). This large-scale and high-resolution dataset may help researchers of many scientific fields, as well as those involved in fishery management and in the update of existing management plans for deep-water red shrimp fisheries as foreseen in relevant General Fisheries Commission for the Mediterranean (GFCM) recommendations.

## 1 Introduction

The continental slope is the transition area between shallow continental shelves and the deep-sea basin. It is usually characterized by marked steepness and cut by submarine canyons that enhance productivity and act as biodiversity refuges (Sardà et al., 2009). As a result of overexploitation of coastal resources and of the increasing power and technology of the vessels, trawl fisheries in the Mediterranean Sea expanded their exploration of the continental slope in the last twenty years, raising concerns for the potential effects of fishing on deep sea ecosystems (Morato et al., 2006). Indeed, most living communities inhabiting these deep areas share characteristics typical of k-type life history - such as slow growth patterns and long-life expectations - that make them particularly vulnerable to anthropic pressure (Tecchio et al., 2013). Endangered deep-sea species include coral communities, some of which of great environmental importance and that can live for centuries if




undisturbed (Bo et al., 2015), and elasmobranch communities that may suffer from by-catch and prey reduction (Barría et al., 2018). In addition to the direct effects of removal and damage of these iconic individuals, deep-sea fishing affects the three-dimensional structure of the sea bottom, modifies the turbidity and attracts scavengers, thus resulting in modifications of the overall community composition (Clark et al., 2016; Roberts, 2002). For these reasons, research aimed to protect deep sea areas

need detailed estimations of the exploited fishing grounds, and these have become available only in recent years (de Juan and Lleonart, 2010).

The deep-sea fishery in the Mediterranean Sea was historically driven by the commercial profitability of deepwater red shrimps (DWRS: *Aristeus antennatus* and *Aristeomorpha foliacea*; Rinelli et al., 2013; Gorelli et al., 2016). Fisheries targeting DWRS can be considered traditional in the western and central areas of the basin (e.g., Relini and Orsi Relini, 1987; Gorelli et al.,

2016), while fleets of the easternmost countries (e.g., Greece and Turkey) started to exploit their own deep waters only around the early 2000's (Politou et al., 2003; D'Onghia et al., 2005; Deval, 2019). Nevertheless, there is evidence that these fishing grounds were not virgin at that time, as they had already been discovered by Italian trawlers (Garofalo et al., 2007; Pinello et al., 2018; Vitale et al., 2014).

The implementation of ship-related self-reporting systems (e.g., VMS: Vessels Monitoring System and AIS: Automatic

Identification System) was a breakthrough to track fishing vessel mobility. In particular, AIS unencrypted radio signals, compulsorily transmitted by European fishing vessels over 15 m to avoid collisions, result in consistent data to observe large trawlers deep-sea targeting shrimps and it has permitted the observation of the supra-national capillary expansion in the eastern regions of the central Mediterranean fleets (Armelloni et al., 2021). Understanding spatio-temporal dynamics of fishing is crucial to prevent stock collapses, especially in deep fishing grounds where several episodes of initial period of high reward

followed by phenomenon of local depletion are documented (Roberts, 2002).

In recent years the General Fisheries Commission for the Mediterranean (GFCM) of the Food and Agriculture Organization (FAO) supported the improvement of knowledge on DWRS stock status and coordinated data collection at Mediterranean level, including a data call on fishing activity (FAO, 2021). The GFCM effort was necessary since the lack of standardized data over the Mediterranean Sea has hindered a comprehensive evaluation of the status of these profitable resources so far.

The need for data collected over a broad scale derives from the biological characteristics of the DWRS, which form only a few genetically distinct populations in the Central and Eastern Mediterranean Sea (Spedicato et al., 2022). Since a mismatch between the stock boundaries and the spatial location of the data collected undermine the accuracy of the stock assessment (e.g., Goethel et al., 2011), the early attempt to evaluate the status of the stock (Ragonese and Bianchini, 1996) was not followed by regular updates. At the time of writing, validated stock assessments exist for the Ionian area, i.e. Geographical

Sub Areas (GSAs) 18 and 19) and the Strait of Sicily (GSAs 12-16) (FAO, 2022; STECF, 2021a), all suggesting that the current level of exploitation may not be sustainable. Nevertheless, the need to continue working on the reconstruction of the exploitation pattern has been repeatedly highlighted (FAO, 2021). The present work (also conceived in the context of FAO, 2021) is motivated by the need for an in-depth understanding of the spatio-temporal distribution of the fishing grounds for DWRS. Mapping fishing grounds is indeed necessary for a correct interpretation of fishery-dependent data, especially in cases



of high fleet mobility, such as this one. A fleet may exploit a fishing ground until causing local depletion and then move to an unexploited one, keeping the catch rates steadily high (Walters, 2003). Therefore, the influence of fishing exploitation dynamics on catchability is a real concern (Paloheimo and Dickie, 1964) that can lead to stock collapses if not accurately accounted for (Ducharme-Barth and Ahrens, 2017).

Here we provide fishing effort data of bottom trawlers exploiting the fishing grounds located between 400 and 800 m, where
DWRS concentrate. Data are provided as monthly estimates at detailed spatial resolution (~ 1 km2) over 4 years (2015-2018). We envisage that the trawling effort data provided here will be of great use for researchers who seek to understand the impacts of deep-water fishing on the targeted stocks, as well as the interactions with deep water community (e.g., bycatch) and/or the economic factors that drive such wide fleet mobility.

## 2 Methods

Fishing pressure exerted by bottom otter trawlers was estimated using AIS data over a time period of four years (2015-2018) in terms of monthly fishing hours and frequency of trawling disturbance.

The study area covers 15 GFCM GSAs in the eastern-central Mediterranean Sea (GSAs 12-16 and 18-27). Based on the need to investigate the main deep-water fishing grounds, focus was placed on the deep-water (DW) bathymetric stratum between 400 and 800 m and intersecting the study area (hereafter named DW stratum) and the fleet was subset in order to retain only
trawlers that most frequently fish within it.

Specifically, analysis was mostly performed following the approach described in Coro at al. 2022 and applying the R scripts available in the R4AIS code repository Version v1.0.2 (Galdelli et al., 2021 ; https://github.com/MAPSirbim/AIS_data_processing/tree/v1.0.2) in order to: (i) reconstruct individual fishing trips, (ii) classify them on a monthly basis according predefined gear classes, and (iii) extract fishing operations. Released data were
gridded at 100th degree resolution. Grid cells were linked to the GFCM rectangles (GFCM Statistical grid, 2022), measuring 0.5° by 0.5° and identified by a 5-digit code (GFCM_COD, representing latitude and longitude by a mixed letter and number code). In particular, each 1 km cell inherited the GFCM_COD of the rectangle intersecting its centroid, as well the GSA of belonging.

The work was carried out in the framework of the FAO GFCM deep water red shrimp Working Group (FAO, 2021), which
supported a further understanding of the specific fishery and validated outcomes.

### 2.1 Data collection

The analysis was based on terrestrial AIS 2015-2018, owned by CNR-IRBIM and obtained from a private provider (http://www.astrapaging.com/), with a poll frequency of 5 min, including EU and non-EU vessels (AIS type = 30).

Data are hosted in a local database that is continuously maintained and updated by the owner. The structure of the database,
as well as the information stored in it, reflects the application of the R4AIS workflow (Galdelli et al., 2021). Historical raw

signals were imported as spatial features (i.e., point geometry data type) in vessel-specific tables. Processing outcomes, such as reconstructed trips and estimated fishing segments, are instead stored as linestring geometries in related vessel-specific tables.

## 2.1 Data processing

The effort dataset provided by the present work derives from fishing vessels that were classified as trawlers and that exploited the DW stratum for a substantial amount of their fishing time. To identify candidate fishing vessels, we first queried the database to select fishing tracks of trawlers that were recorded at least once a year in the DW stratum. The trawling activity in each $0.01° \times 0.01°$ grid cell was then estimated by: (i) intersecting retrieved fishing tracks with the grid, (ii) computing the durations of each fishing portion overlapping the cell (i.e., dividing resulting length by the inherited mean fishing speed that

is stored in the fishing segments' table), (iii) aggregating fishing hours (Fh) by month and vessel. To further refine the composition of the selected fleet, fishing hours of each vessel were summed by year and depth strata (DW: deep water and SW: shallow water, respectively) and used to apply two subsequent filters. These excluded vessels spending less than a predefined threshold in DW (filter #1 in Appendix A, Fig. A1) and vessels whose $Fh_{DW}/(Fh_{DW}+Fh_{SW})$ ratio was less than a predefined threshold (filter #2 in the Appendix A, Fig A2). For both filters, different threshold values were tested to balance

between the need to remove "noise" (trawlers that rarely performed deep-sea fishing activities or just fished close to 400m depth contour) and the intent to minimize the loss of fishing activity exerted in the reference depth stratum.

Lastly, fishing hours were preliminary aggregated by cell id (FID) at the annual level to conduct a sensitivity analysis and exclude those cells that contribute negligibly to DW fishing activity. Resulting data were used to estimate the monthly fishing effort in all the cells of the study area, also beyond the DW stratum. Then, following the method proposed by Amoroso et al.,

(2018), we also estimated the trawling frequency of the cells within the DW stratum as average days between trawling events over the whole time series.

## 3 Results

The first query to the database returned the activity of 614 trawlers (gear type OTB and OTB2, see R4AIS workflow) fishing at least once in the DW of the study area. Then, for each year, we filtered out those vessels fishing less than 20 hours in DW

(vertical dotted line, Figure A1). This filter reduced the number of vessels by 30% (420 vessels), causing a decrease in the fishing activity of around 24% in the entire area and of 0.87% in the DW stratum. Afterwards, we excluded those vessels spending less than 5% (0.05) (vertical dotted line, Figure A2) of their fishing activity in DW fishing grounds. The application of this second filter decreased fishing effort by 19% in the entire area and by 2% in DW stratum. The final dataset comprised 370 vessels undertaking 48,814 trips and producing fishing tracks for a total duration of 961,702 hours (Table 1). Out of these,

37% were spent exploiting DW fishing grounds.



**Table 1: Summary description of the AIS validated dataset resulting from the filtering procedure**

| Year | Vessels | Fishing trips | Fishing hours | DW Fishing hours | DW Fishing hours/Total Fishing hours (%) |
|---|---|---|---|---|---|
| **2015** | 208 | 11988 | 213448 | 75613 | 35 |
| **2016** | 242 | 12949 | 251848 | 86222 | 34 |
| **2017** | 236 | 12544 | 256977 | 94557 | 37 |
| **2018** | 227 | 11333 | 239429 | 95631 | 40 |
| **Total** | 370 | 48814 | 961702 | 352024 | 37 |

Following a conservative approach, we finally adopted a threshold of 0.33 hours (per cell per year) to exclude grid cells that
contributed negligibly to DW fishing activity. Therefore, grid cells with less than 0.33 hours per year were discarded, causing
a decrease of about 1% in the total DW fishing hours over the whole study period (Table 2).

**Table 2: Sensitivity analysis and alternative thresholds tested, before identifying the validated dataset.**

| Threshold (annual hours in a cell) | Total DW fishing activity (hr) | Proportion of DW activity retained |
|---|---|---|
| 0 | 352023.7 | 1.00 |
| 0.33 | 348254.3 | 0.99 |
| 0.66 | 342738.9 | 0.97 |
| 1 | 336974.7 | 0.96 |

From the cumulative pattern of fishing activity (in hours of fishing/km2) we observed that most of DW trawling fishery is
concentrated in the Strait of Sicily (GSAs 12, 13 15 and 16), while other grounds are locally exploited in the Southern Adriatic
Sea (GSA 18), and offshore Crete, Greece and Turkey (Fig. 1). By contrast, an extremely low number of fishing vessels
broadcasted AIS in DW areas offshore Cyprus and North African countries, and almost no deep-water operations were reported
in the eastern-southern parts of the Mediterranean Sea.


## Cumulative fishing effort (2015-2018)

Figure 1. Cumulative fishing hours 2015-2018 within the DW stratum (light grey polygon) in the Central-Eastern Mediterranean Sea. The borders of the management units (GSA) are reported in black.

The monthly time series of fishing hours highlights a clear seasonal trend, with DW trawlers being more active in spring/summer (Q2 and Q3, respectively) than in fall/winter (Q4 and Q1, Fig. 2). This seasonality affects the intensity more than the spatial extent of fishing effort, especially in those areas characterized by very concentrated fishing grounds. Such is the case of Crete (GSA 23), whose northern narrow and precise fishing grounds were persistent along the year, but exploited during spring/summer with relatively higher intensity. On the contrary the exploitation of its southern grounds was mostly
confined in the second half of the year (Fig. 2).



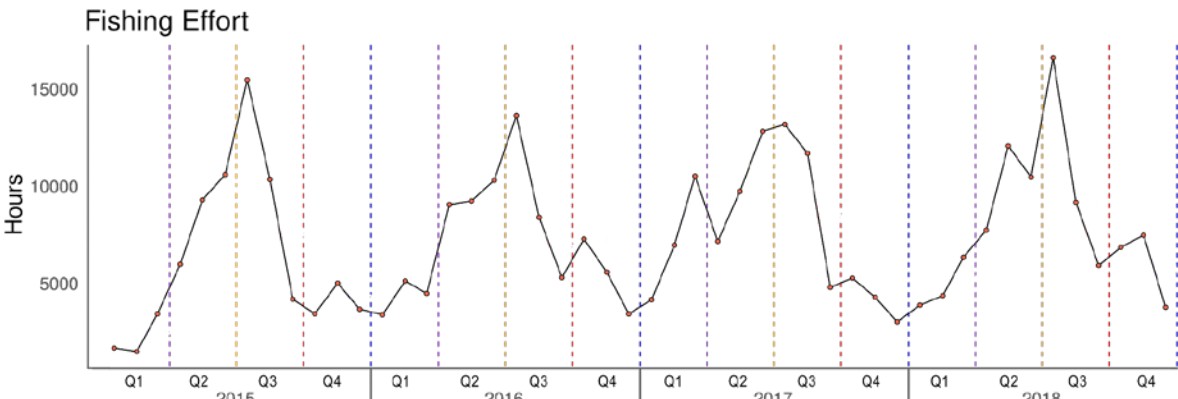

**Figure 2: Monthly value (red dots) of total fishing hours within the DW stratum of the Central-Eastern Mediterranean Sea for the period 2015-2018. Seasonal limits (vertical dotted lines) are located at the starting day of each quarter (Q).**


The spatial distribution of trawling frequency was consistent with the pressure metrics, highlighting that DW areas identified as those where fishing activity aggregated the most (e.g., the Strait of Sicily) were fished with the highest frequencies (Fig. 3). These cells were mostly on or close to the 400 m depth contour, and close to this depth between the Sicilian grabens. The overall average interval between fishing events in the whole study area was 164 days (about 5 months and half).


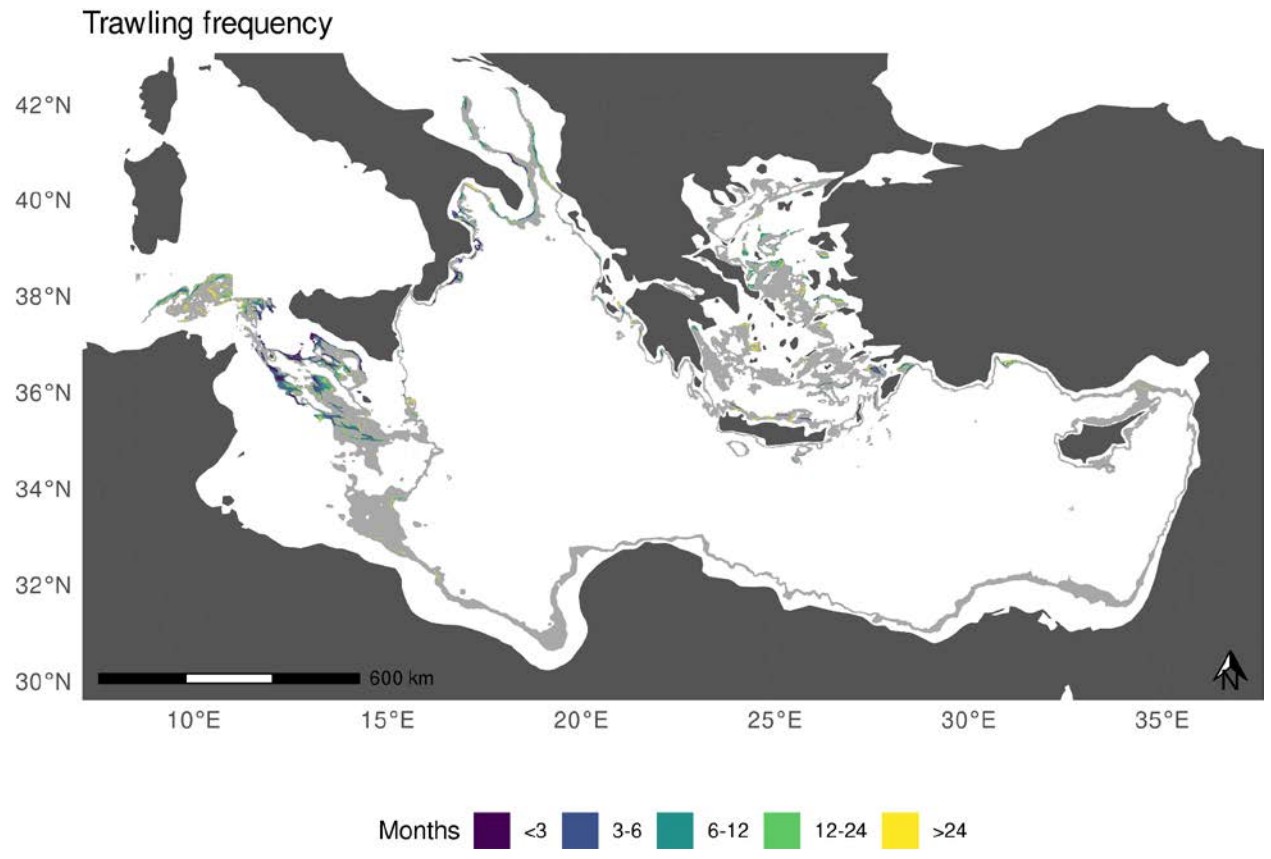

**Figure 3. Frequency of deep-water trawling events at depths of 400-800 m (light grey polygon), years 2015-2018**

Focusing again on GSA 23, Fig. 4 confirms that the heavily exploited fishing grounds off the north shore of Crete accounted

for the highest trawling frequency class (less than three months between fishing events), while the southern fishing grounds were exploited more sporadically. Nevertheless, most of the fishing grounds appear as narrow and elongated paths repeatedly exploited over the years (trawling frequencies < 24 months), while only some larger fishing grounds are less visited (frequency over 24 months) and confined in the north area. They probably indicate exploratory fishing.

**Figure 4: Seasonal distribution of fishing effort (up) and mean trawling frequency (down) in DW stratum (light grey polygon) off the island of Crete (GSA 23), years 2015-2018. Effort is mapped in terms of average fishing hours estimated in each quarter Q. Trawling frequency, originally expressed as the mean interval in days between fishing events, was categorized by months.**

## 4 Discussions

This study presents one of the few model-inferred and expert validated reconstructions of DW fishing effort in the Central-Eastern Mediterranean Sea. Released AIS-based data, aggregated at a monthly time step and spanning over a four-year period, describes in detail the DW trawling dynamics in the study area.

Outcomes of the analysis suggest that the DW stratum is mostly fished in its central-eastern part with the exception of the south-east where AIS data are also mostly lacking, and highlight two contrasting pictures. The Strait of Sicily has a peculiar





exploitation pattern and is by far the area where the DW stratum is the most extensively trawled. The observed Sicilian fishing grounds are broad, spanning over dozens of square kilometers in all directions, thanks to the wide extension of the related DW strata but also probably due to the long tradition and specialization in the DW fishery (Pinello et al., 2018). Moving eastwards, DW fishing grounds radically change into elongated patterns and best the examples are in the southern Aegean and off Cyprus, where relatively precise repeated tows concentrate over narrow and very long fishing grounds. Interestingly, eastern

Mediterranean fishing grounds are confined to small areas even where the DW stratum is moderately broad, as is the case off the Turkish coasts.

Comparing these results with those presented in earlier studies focusing on the deep-sea fishery in the central and eastern Mediterranean Sea, similarities may be noted underlining the consistency of the data released. The Strait of Sicily and, to a lesser extent, the Italian Ionian Sea are confirmed as historically exploited fishing grounds (Ragonese and Bianchini, 1996;

Carlucci R et al., 2018), while the scarce information available agrees on a very low exploitation of DW fishing grounds by Greek fisheries until 2010 and thus does not permit to cross-validate our observations (Papakōnstantinou et al., 2007; Kapiris and Thessalou-Legaki, 2011; Guillen et al., 2012). Nevertheless, literature suggests that Italian vessels exploiting eastern fishing grounds are usually based in Crete and Rhode islands (Pinello et al., 2018), where several DW fishing grounds are observed by the present study. Moving to Turkey, available literature documents the existence of a recently established DW

trawl fishery off Antalya Bay where our study maps fishing activity (Deval et al., 2009, 2016). Regarding Cyprus no information is found on the exploitation of local fishing grounds, but the trawling activities detected in the south of the island may, again, be explained by Italian bottom trawlers operating far from their country of origin (Pinello et al., 2018).

One of the major concerns regarding the dataset released here is the lack of observed effort in the African part of the basin as it significantly prevents a consistent picture of DW fishery in the whole study area. In the case of the Mediterranean Sea, which

is relatively small and enclosed, the major well-known data-gaps are due to the small number of northern African vessels equipped with AIS (Merino et al., 2019), and this overshadows all other technical issues related to the signal coverage and the exploitation of far deep-sea fishing areas (Ferrà et al., 2020; Holmes et al., 2020). In light of the above, AIS-observed DW grounds in Mediterranean southern areas - including the data released in the present work - are inevitably underestimated in terms of fishing pressure, or even lost. Importantly, this prevents the adequate monitoring of those grounds where DW fishing

has recently developed (i.e., Egypt; Farrag, 2016). Future work should concentrate on integrating available data with additional complementary data sources such as VMS, non-cooperative Synthetic Aperture Radar or optical imagery (Global Fishing Watch - A Radar-Illuminated Ocean Reveals Dark Fleets, 2022), as well as alternative methodological approaches that combine geographical data and expert knowledge to estimate fishing pressure (Kavadas et al., 2015; Maina et al., 2016).

Explicit fishing effort datasets covering the entire Mediterranean are scarce, limiting possibilities of comparing the present

study with other sources of information. The only resource on trawling frequency available in the literature - in addition to the data presented herein - is the one released by Amoroso et al. (2018), in which the mentioned metric covered a smaller area of the Eastern-Central Mediterranean Sea (i.e., Aegean Sea) over the period 2008-2010. Trawling frequency was found as one of the most relevant in the assessment of trawling impacts on the seabed biota (Amoroso et al., 2018).



Above all, the flagship dataset on apparent fishing effort made freely available by Global Fishing Watch (GFW; Kroodsma et al., 2018) through its marine manager portal is worthy of mention. Fishing fleets (i.e,, vessels with the same fishing gear) are inferred using a deep learning model trained on a sample of vessels with gears retrieved (and in part manually reviewed) from official registries (Kroodsma et al., 2018). On the contrary, we classified single trips based on gear-specific rules that were formulated by experts working with a sample of vessels operating known gear. Even though a pure model-based approach is optimal for large scale aggregations (Coro, 2020), it may have a worse performance at finer resolution such as that of the present study. Thus, the application of a mixed approach combining model inference and expert validating rules, such the R4AIS method used here, could be in our opinion preferable for local application at high spatial resolution.

Another spatially explicit dataset on DW trawling is provided by the Fishery Dependent Information (FDI) data call of the European Union (STECF, 2021b). It represents the only official dataset disaggregated at fishery and area level, and the best suited to validate the DW shrimp fishery in the study area as it includes target species as a filter option (Gibin et al., 2021). Nevertheless, statistics are limited to European vessels and reported at coarser spatial and temporal resolutions (i.e., fishing day gridded at 50 km resolution), preventing a direct comparison with the dataset released here.

Last but not least, an additional, important, source of official information is the VMS, which has been mandatory on EU fishing vessels since the early 2000s (e.g., Lee et al., 2010). If VMS could be key to run comparisons and integrate data gaps, not all countries in the Mediterranean region currently operate a VMS-based fishing monitoring centre and data access has historically been restricted to government regulators or other fisheries authorities. Even though an attempt was made in the framework of the EMODnet MedSea Checkpoint (Martín Míguez et al., 2019; Tassetti et al., 2016), VMS-observed trawling effort has not yet been successfully reconstructed at Mediterranean basin level.

## 5 Data availability

Available deep-water trawling effort data are publicly accessible on the SEANOE repository (https://doi.org/10.17882/89150, Pulcinella et al., 2022), in terms of: (i) monthly hours vessels spent operating trawl gear, (ii) number of vessels performing those activities and (iii) average interval in days between trawling events. Metrics are coherently gridded at 0.01° - using only cells intersecting the 400m-800m depth stratum of the Eastern-Central Mediterranean Sea (GSAs 12-16 and 18-27) - but released as separate files for a better management of their table schemas. Fishing hours per cell (and related number of vessels exerting this effort) are indeed available on a monthly basis, and easier to handle by adding the field "year". On the contrary, a single mean trawling frequency was estimated in each cell over the full time series, irrespective of the year.

The explicit link with the GFCM statistical grid (by the field "GFCM_COD") promotes future joining/aggregation with sources of official data.

The spatially explicit dataset provided is believed to be reliable to describe spatial and temporal patterns of deep-water trawling fishery in the Eastern-Central Mediterranean Sea. The dataset is accompanied by discovery metadata describing the main data

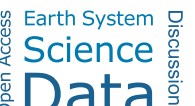

caveats and gaps, and the R scripts used to reconstruct trawling effort. Indeed, it is the responsibility of the data user to take

this information into account when reusing data and outcomes could be reproduced with own AIS transmissions.

The dataset is released under the Creative Commons Attribution license (CC-BY, v. 4.0, https://creativecommons.org/licenses/by/4.0/deed.it, last access: 8 August 2022) and it follows the FAIR principle of Findability, Accessibility, Interoperability and Reusability of data (Wilkinson et al., 2016).


## 6 Code availability

The R code used to process AIS data was made available by Galdelli et al., 2021 at: https://github.com/MAPSirbim/AIS_data_processing, while the scripts required to reproduce the plots reported in the present work are available within the data repository itself (Pulcinella et al., 2022).

Additional routines used for the processing and classification of the AIS data are available at: https://github.com/irbimMAPS/dwrs_workshop_2022.

## 7 Conclusions

A four-year monthly time series of AIS-based observed patterns in deep-water bottom contact fisheries in the eastern-central

Mediterranean Sea is presented (grid resolution: 1 km), accounting for the spatial extent and temporal variability in fishing activity during the period 2015-2018. Released data depict deep-water trawling in terms of its dynamics, spatial distribution and seasonality, and was envisioned to support in fulfilling the requirements of Recommendations GFCM/42/2018/3, GFCM/42/2018/4 and GFCM/43/2019/6 on a multiannual management plan for sustainable trawl fisheries targeting giant red shrimp and blue and red shrimp in the Levant Sea (GSA 24, 25, 26 and 27), the Ionian Sea (GSA 19, 20 and 21) and the Strait

of Sicily (GSA 12, 13, 14, 15 and 16), respectively, which include the provision of maps of fishing grounds. Given the fact that deep-sea fishing vessels are mostly large enough to adopt the AIS, the released data were considered to be highly representative for this study.

Released data could be used to address other issues. Among them, we should mention the understanding of the ecosystem that is impacted by this fishery. Indeed, even though a very small portion of the deep-water seabed is trawled, information on

existing biocenosis should be overlaid to infer and quantify the impact on this seabed. Also, the different degree of seasonality should be investigated, as well as its main drivers (e.g., resource availability or just market prices and fuel costs?). These might suggest potential spatial conflicts between fishers and have management implications. Lastly, released data, if linked to logbook-based information as well as to a sample of logbook catches, may help in solving the dilemma of catch allocation and origin - a crucial missing piece of information for the assessment and management of long-ranging fisheries such as the DWRS

fishery in the eastern-central Mediterranean.

We encourage the use of the spatio-temporal dataset on deep water trawling effort provided here to all those caring for deep ecosystem conservation and sustainability of marine living resources, both of which are advised by a better understanding of the impacts induced by anthropogenic pressure.





## Author contribution

JP, ENA, CF, ANT conceived the research idea. ANT, ENA, JP and CF developed the methodology to reconstruct fishing effort and contributed to the collection and curation of data described in this paper. JP and ENA produced the results. JP, ENA and ANT wrote the original draft. CF and GS reviewed and edited the manuscript. ANT supervised the work. All authors participated in the interpretation of results and gave final approval for publication.

## Competing interests

The authors declare that they have no conflict of interest.

## Acknowledgements

We are grateful to the GFCM Deep Water Red Shrimp Working Group for its help in preparing and discussing the reconstructed trawling effort data.

## Financial support

This research has been supported by a letter of agreement between the Food and Agriculture Organization of the United Nations (FAO) under the General Fisheries Commission for the Mediterranean (GFCM) and CNR-IRBIM for the provision of *"Identification and mapping of bottom trawl fishing grounds for deepwater red shrimp in the Eastern-Central Mediterranean Sea (GSAs 12-16, 18-27)"*.

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



**Appendix A**

In order to identify the most representative deep-water trawling fleet, two subsequent filters considered: (i) the annual fishing
activity exerted by the vessel within the 400-800m depth stratum and (ii) the proportion between its deep-sea fishing activity
and the total activity (both in shallow and deep fishing grounds). In detail:

- Filter#1: DW fishing hours ≥ 20. On a yearly basis, vessels fishing little within the DW stratum were filtered out. The
  threshold of 20 fishing hours in DW (vertical dotted line, Figure A1) was considered appropriate as the DW fishing
  activity decreased only by 0.87% during the whole study period, while the global fishing pattern (total fishing hours
within and outside DW fishing grounds) by 24%. Filter#1 reduced the fleet from 614 to 420 vessels (~31%).
- Filter#2: DW fishing hours/ Total fishing hours ≥ 0.05. On a yearly basis, vessels fishing little within the deep-water
  stratum compared to what they did outside were filtered out. Applying the threshold of 0.05 (vertical dotted line,
  Figure A2), fishing activity in DW fishing grounds decreased only by 2%, while in the global fishing activity (both
  in shallow and deep fishing grounds) there was a reduction of 19%. Filter#2 reduced the fleet from 420 to 370 vessels.


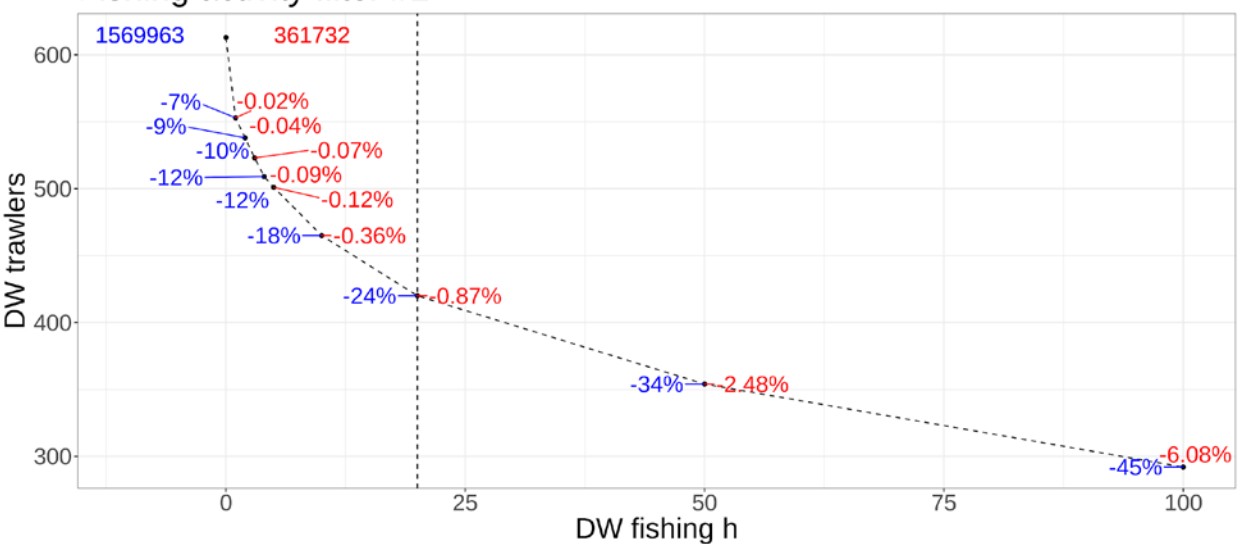

**Figure A1: Overall decrease (%) in the cumulative fishing hours within the DW fishing grounds (red labels) and within the whole
Mediterranean (blue labels), by filtering out vessels (Y axis) DW fishing less than a number of hours per year (X axis). Top labels
show the overall initial fishing hours (over the whole study period), while the vertical dotted line represents the chosen threshold (20
hours).**

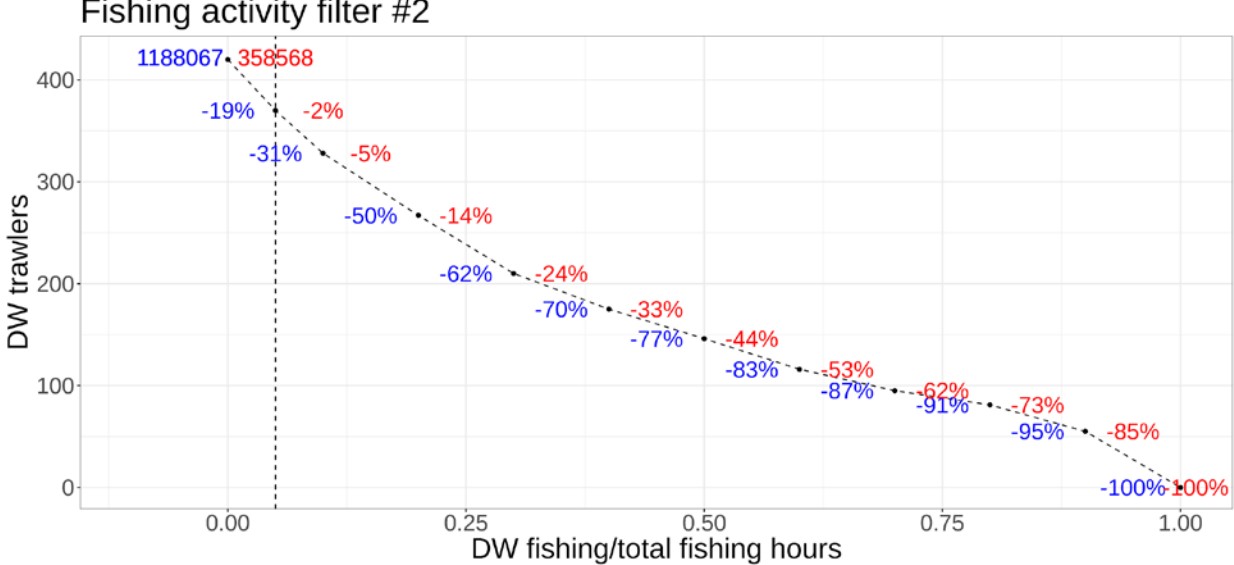

**Figure A2: Overall decrease (%) in the cumulative fishing hours within the DW fishing grounds (red labels) and within the whole Mediterranean (blue labels), by filtering out vessels (Y axis) fishing in DW less than a percentage compared to their total effort (X axis). Top labels show the overall initial fishing hours (over the whole study period), while the vertical dotted line represents the chosen threshold (0.05).**
