# Peer review of "Deep-water red shrimp fishery in the Eastern-Central Mediterranean Sea: AIS-observed monthly fishing effort and frequency over four years"

_Earth System Science Data, 2022_

## Author Response (AR1)

| COMMENTS FROM REFEREES | AUTHOR'S RESPONSE |
|---|---|
| **RC1: 'Comment on essd-2022-318**
**https://doi.org/10.5194/essd-2022-318-RC1**
The article is worth of publication since the work done is relevant, both from a scientific and technical point of view. Moreover, the value of the work is increased by the sharing not only of dataset but also of meaningful metadata and the code. Also, explanation of the work is clear and the method and results are presented in a consistent and effective way. I would only suggest some minor technical corrections that are:

Row #35: I noticed that the bibliography seems a bit outdated. There's nothing more recent to cite on this theme? | Row #35: The bibliography was intentionally bit outdated as it refers to the date on which vessel positional data became available.
**AUTHOR'S CHANGES IN MANUSCRIPT:** Anyway, we added a more recent reference and edited the text into "available only in the last decade" |
| Row #71: I suggest to use "valuable" instead of "of great use"
Row #87-88: I suggest to turn "of belonging" at the end of the sentence in "it belongs to"
Row #108: I suggest to specify the metric of threshold using for example "predefined temporal threshold" in place of "predefined threshold"
Row #109: similarly, I suggest to use "predefined nondimensional threshold", also to differentiate the two-threshold named
Figure 1: it is a pity that fishing hours distinctions is not clearly visible from the figure. Maybe adding some zooms on above mentioned areas (Strait of Sicily, Southern Adriatic and Offshore Crete) could help the figure being more meaningful?
Figure 2: add the "Years" (or "Quarters", maybe) label to the horizontal axe
Row #206: I would put the name of the cited web article in quotes: "Global Fishing Watch - A Radar-Illuminated Ocean Reveals Dark Fleets", 2022
Row #215: there's a double comma instead of a point and comma in "i.e., vessels..."
Row #244: I would turn "discovery" into "machine-readable" when speaking about metadata | We made all the remaining suggested minor corrections.
**AUTHOR'S CHANGES IN MANUSCRIPT:** Given the nature of the issues raised, these were revised directly in the manuscript.

Fig.1 and Fig.2 were modified accordingly to the received suggestions.
**AUTHOR'S CHANGES IN MANUSCRIPT:** Fig.1 and Fig.2 were replaced |
| **RC2: 'Comment on essd-2022-318'**
**https://doi.org/10.5194/essd-2022-318-RC2**
The date sets of the Vessels Monitoring Systems are the great mine of information on fishing fleet mobility and the estimation of the catch and discards. Therefore, the presented work is valuable, technically and scientifically adequate for the publication. The problem of the lack of clarity in the details in the figures is important. However, since the previous reviewer also detected the same problem, the authors reported that the | Correct, we tried to make Fig.1 and Fig.2 with higher quality and more meaningful after RC1 comments |

| figures were corrected. Journal's readers will be much happier with figures with higher quality and more meaningful.

Fishing Vessels Monitoring System (BAGÄ°S: BalÄ±kçÄ± Gemileri Ä°zleme Sistemi) of the Turkish vessels use three different data communication types: Gsm, AIS and Satellite (I am sending a small dataset attached). In this case, calculation of trawling effort of the Turkish Fishing vessels using only AIS-based data sets may not give real result. What can the authors say about this problem? | We actually observed more gaps in Turkish AIS data especially after 2016 (when BAGIS entered into force), without decreasing the number of observed equipped MMSI/vessels. To give an example in the figure below we plotted some AIS samples before and after 2016 (see below in this document).
As pointed out by Reviewer 2, it results in effort underestimates and it could be due to BAGIS official traceability system broadcasting trough different data communication types (e.g., GSM networks).
Nevertheless, the AIS analysed data remains – unfortunately - the only freely available.
**AUTHOR'S CHANGES IN MANUSCRIPT:**
We edited the manuscript accordingly (lines 206-208) highlighting this underestimation together with the one observed in the Mediterranean southern areas. In particular, we added *"Underestimation of DW fishing effort also applies to the eastern Turkish grounds where available AIS pings could be incomplete due to official traceability systems broadcasting trough different data communication type (i.e., BAGIS using Gsm, AIS and Satellite from year 2016; Tolon, 2017)."* |
|---|---|

[Figure]

AIS broadcasts of sample Turkish vessels in 2015 and then in 2019 (after 2016, when BAGIS entered into force).
(a) AIS broadcasts of a vessel during the same month in 2015 and in 2019, (b) pings broadcasted by a trawler, fishing in the same areas reported by the sample given by Reviewer 2, in 2015 and 2019.